# Mannosylated Polymeric Ligands for Targeted Delivery of Antibacterials and Their Adjuvants to Macrophages for the Enhancement of the Drug Efficiency

**DOI:** 10.3390/ph15101172

**Published:** 2022-09-21

**Authors:** Igor D. Zlotnikov, Alexander A. Ezhov, Rostislav A. Petrov, Maksim A. Vigovskiy, Olga A. Grigorieva, Natalya G. Belogurova, Elena V. Kudryashova

**Affiliations:** 1Faculty of Chemistry, Lomonosov Moscow State University, Leninskie Gory, 1/3, 119991 Moscow, Russia; 2Faculty of Physics, Lomonosov Moscow State University, Leninskie Gory, 1/2, 119991 Moscow, Russia; 3Institute for Regenerative Medicine, Medical Research and Education Center, Lomonosov Moscow State University, 27/10, Lomonosovsky Ave., 119192 Moscow, Russia; 4Faculty of Medicine, Lomonosov Moscow State University, 27/1, Lomonosovsky Ave., 119192 Moscow, Russia

**Keywords:** drug resistance, efflux inhibitor, CLSM, moxifloxacin, eugenol, macrophage, flow cytometry

## Abstract

Bacterial infections and especially resistant strains of pathogens localized in macrophages and granulomas are intractable diseases that pose a threat to millions of people. In this paper, the theoretical and experimental foundations for solving this problem are proposed due to two key aspects. The first is the use of a three-component polymer system for delivering fluoroquinolones to macrophages due to high-affinity interaction with mannose receptors (CD206). Cytometry assay determined that 95.5% macrophage-like cells were FITC-positive after adding high-affine to CD206 trimannoside conjugate HPCD-PEI1.8-triMan, and 61.7% were FITC-positive after adding medium-affine ligand with linear mannose label HPCD-PEI1.8-Man. The second aspect is the use of adjuvants, which are synergists for antibiotics. Using FTIR and NMR spectroscopy, it was shown that molecular containers, namely mannosylated polyethyleneimines (PEIs) and cyclodextrins (CDs), load moxifloxacin (MF) with dissociation constants of the order of 10^−4^–10^−6^ M; moreover, due to prolonged release and adsorption on the cell membrane, they enhance the effect of MF. Using CLSM, it was shown that eugenol (EG) increases the penetration of doxorubicin (Dox) into cells by an order of magnitude due to the creation of defects in the bacterial wall and the inhibition of efflux proteins. Fluorescence spectroscopy showed that 0.5% EG penetrates into bacteria and inhibits efflux proteins, which makes it possible to increase the maximum concentration of the antibiotic by 60% and maintain it for several hours until the pathogens are completely neutralized. Regulation of efflux is a possible way to overcome multiple drug resistance of both pathogens and cancer cells.

## 1. Introduction

To combat dangerous infections, pathogens resistant to many drugs, and dormant bacteria, it is necessary to create new combined drugs [1,2]. The key direction is targeted drug delivery to increase organ bioavailability, active concentration of the drug, and its duration of action [3,4,5,6,7,8]. Additional strengthening of the antibiotic can be achieved through the use of substances called efflux inhibitors [9,10,11,12,13,14]. Currently, antibiotic-resistant pathogens [15], especially those localized in macrophages or granulomas [16], are an acute problem. Targeted delivery of effective fluoroquinolones, inhibitors of bacterial DNA gyrase, by means of mannosylated molecular containers discussed above will allow the drug to be localized in the focus of infection. 

On the other hand, antibiotic resistance can be defeated by delivering an adjuvant substance. It is known from the literature that the active efflux of both prokaryotic and eukaryotic cells strongly reduces the efficiency of a large number of antibiotics, and efflux greatly determines the effect of multidrug resistance [12,13]. The study of efflux is a rather complex process; the described techniques are based on the reduction of MIC of model efflux inhibitors (i.e., ethidium bromide, chlorpromazine) with the addition of mainly essential extracts (Thai herbs, allylbenzenes, i.e., EG [11,17,18,19]. In this paper, we present a new technique for studying efflux using CLSM. The development and use of specific, powerful efflux inhibitors will significantly increase the effectiveness of drug formulations and prevent the development of pathogen resistance.

Previously, a significant potential of allylbenzenes, including eugenol (EG), and terpenoids, which are synergists of fluoroquinolones, has been demonstrated [10,17,20,21,22,23,24,25,26]. As promising adjuvants, apiol and its analogs dillapiole, myristicin, and nothoapiol should be noted [27,28]. Some monoterpenes, namely (–)-carveol, geraniol, and citronellol, show a synergy effect when used as additives for conventional antimicrobials (chloramphenicol, minocycline, amoxicillin, and ciprofloxacin) [13,14]. These substances are capable of creating defects in the bacterial membrane [17,23] and probably inhibit enzymes and efflux pumps [9,10,11,12,13,29] of pathogens that cause resistance of strains. Thus, the complex preparation of the fluoroquinolone + adjuvant combination with the function of active targeting of mannose receptors of macrophages is a promising method for the treatment of resistant bacterial infections.

Often, resistant bacteria are localized in macrophages or granulomas [16,30,31,32,33]. Macrophages play an important role in both innate and acquired (humoral and cellular) immune responses [34]. The effect on macrophages can be classified into three groups according to the mechanism: activation of macrophages from inactive in M1 (pro-inflammatory) or M2 (pro-tumor or anti-inflammatory) [35]; repolarization of M2 in M1 to reduce the rapid response of the immune system [36,37]; and accumulation of antibacterial substances in macrophages against microorganisms “hidden in the reservoir” [30]. In this work, the focus is on the delivery of antibacterial drugs to alveolar macrophages for the destruction of resistant microorganisms (including *M. tuberculosis* localized in granulomas). The proposed strategy is based on high-affinity binding to CD206 mannose receptors overexpressed on M2 macrophages [38,39,40,41]. The use of mannose receptors that recognize Man-, Fuc-, and GlcNAc-residues of oligosaccharides of the cell wall of pathogenic microorganisms has a number of advantages in comparison with the strategy of targeting other receptors: exposure only to pathogenic macrophages, reduced risk of resistance and complications, and increased efficiency (https://www.macrophagetx.com; (accessed on 1 September 2022). CD206 is a C-type lectin (175 kDa), which is a transmembrane protein that binds carbohydrates due to a lectin-like region of eight domains [39,42]. However, since CD206 itself is difficult to isolate, model receptor proteins, for example, concanavalin A (ConA), are often used to screen the affinity of mannosylated delivery systems to macrophage mannose receptors [25,43,44,45,46,47,48]. The relevance of the ConA model was previously demonstrated using molecular dynamics and neural network analysis [43] as well as in vivo and in vitro experiments, which demonstrated that highly ConA-specific delivery systems effectively target macrophages [49,50,51].

Thus, using a model receptor, the selection of the greatest affinity for the mannose receptor conjugates and optimization of the structure of the drug delivery system to the macrophage were carried out [25]. The effect of enhancing drug penetration due to delivery systems adsorbed on bacterial wall surfaces was demonstrated [25,52]. The literature describes molecular containers based on liposomes [53,54], cyclodextrins [55,56], oligo- and polyamines [25,57], polyglycols [58,59,60], chitosan [44,61], and mannan [45,60,61]. Synthesis of molecular containers includes two main parts: container synthesis (carrier molecule) and mannosylation (introduction of a mannose label). The main reactions used in the first stage are as follows: imine (Schiff) reaction between carbonyl (i.e., −C=O groups can be obtained by periodate oxidation of carbohydrates) and amino groups with subsequent reduction of the −C=N bond by borohydride [62]; preparation of carboxymethylated derivatives of carbohydrates by exposure to chlorine acetic and hydrochloric acids [63]; activation of hydroxyl groups with TsCl, substitution for an easily leaving group (often iodide), followed by the introduction of a spacer [56,64]; glutaraldehyde, glyoxal, polyamines (spermine, spermidine), carbodiimide, and diisothiocyanates can be used as crosslinking agents [65,66]; and activation of hydroxyl groups with carbonyl diimidazole followed by attachment to the activated carbonyl group of a nucleophilic agent [25]. The easiest way to perform mannosylation is conjugation of the linear form of mannose formed from cyclic in solution with a delivery system with primary amino groups (reductive amination) [25,44,67]. The introduction of the cyclic form of mannose requires either a preliminary modification of the -OH group or the use of commercially available activated forms, for example, with −N=C=S, −NH_2_, −N_3_, and −C=O groups [68,69]. High-affinity conjugates with synthetic analogues of oligomannosides can be obtained using available mannose on spacers -O-CH_2_CH_2_S-Man [56] or on spermine/spermidine [25,44]. It is much more difficult to introduce oligomannosides directly into the conjugate: a multi-stage synthesis is required, which reduces the yield [70]. However, it is possible to use a carbohydrate with a reducing end and perform effective mannosylation, as in the case of simple mannose [71]; this approach is used in this work.

The aim of this work is to develop a combined formulation based on 4th-generation fluoroquinolone moxifloxacin (MF) loaded into mannosylated containers as a target delivery system and enhanced with adjuvants (EG [10,17,18,19,22,72,73,74,75,76,77,78,79,80], menthol, and safrole [23,28,81]) capable of bypassing the resistance of pathogens and fighting dormant infection.

The approach proposed provides a comprehensive, versatile focus on the problem of pathogen resistance. The above-discussed works were aimed either at creating delivery systems (mannosylated particles) or studying the antibacterial activity of antibiotics or substances from the class of terpenoids, which are practically not soluble in aqueous solution, and thus, they have low efficiency themselves. The advantage of the suggested systems that will ensure their therapeutic potential is the combination of several fronts in the fight against resistance, namely the accumulation of drugs in macrophages with simultaneous enhancing of the antibiotic efficiency due to increased permeability of the bacterial cell membrane and inhibition of efflux protein provided by adjuvants.

## 2. Results and Discussion

### 2.1. Synthesis and Physico-Chemical Characteristics of Mannosylated Conjugates

#### 2.1.1. FTIR Spectroscopy

The reactions underlying the synthesis of conjugates 1–3 are as follows (Figure 1): reductive amination of aldehyde groups [67,68,69,71] and activation of hydroxyl groups of cyclodextrin CDI followed by crosslinking with NH_2_– and OH– groups [25,65,82,83]. The lightest conjugate 1 based on amCD and triMan was synthesized according to the scheme in Figure 1a. In the FTIR spectra of amCD-triMan, in comparison with initial compounds (Figure 2a), changes in the region of 2800–3000 cm^−1^ were observed, corresponding to the valence vibrations of C-H and N-H bonds; the intensity of the peak of 1420 cm^−1^ increased, corresponding to the deformation vibrations of N-H at the junction of two fragments: trimannoside and cyclodextrin. Bright changes were recorded in the area of 900–1100 cm^−1^, which corresponds to the most effective splitting of the reducing end of the saccharide and crosslinking with the formation of C-N bonds (Figure 1a). Ligands 2a and 3a (Figure 1b) are polymers with attached trimannoside (triMan) as labels to the mannose receptors of macrophages. triMan-(GlcNAc)_2_ reacts with PEI amino groups via the reducing end to form a Schiff base PEI-N=CH-triMan, which is reflected in the IR spectra (Figure 2b and Appendix A): the area of peaks corresponding to N-H and C-H bending decreases (1300–1550 cm^−1^), and the absorption peak of the bond C=N appears. Grafting of polymer chains with cyclodextrins will allow to shield the charge of PEI amino groups due to its modification and the formation of a protective carbohydrate coat and increase the theoretical drug-capacity since CD are one of the lightest but at the same time effective containers for medicines (ligand 2b, 3b). The activation of secondary hydroxyl groups of cyclodextrin with CDI followed by cyclodextrins’ modification with the primary amines of spermine or PEI leads to final grafted conjugates 2b and 3b (Figure 1b).

#### 2.1.2. NMR Spectroscopy

The obtained data on the structure of the synthesized conjugates were confirmed by NMR spectroscopy. The ^1^H NMR spectra of triMan-GlcNAc_2_, PEI1.8-triMan, and HPCD- PEI1.8-triMan with proton assignment are presented in Figure 3. The signals in the range of 3.9–3.1 ppm and in the range of 5.1–5.3 ppm (Figure 3a) correspond to H2–H5 and H1 of D-mannose or D-glucose (in HPCD) units, respectively [84]. The signals of the anomeric protons H1 lie in a weaker field of 4.6–5.3 ppm [85]. The H2 patterns are shifts of 3.8–3.9 ppm, and for neutral protons, the signals overlap in the region of 3.1–3.8 ppm. The presence of PEI in ligand **2a** (Figure 3b) and the formation of bonds triMan–CH_2_–NH–PEI in conjugates is confirmed by the signals in the range of 2.5–2.9 ppm according to the literature data for mannosylated chitosan grafted with PEI, PEI copolymers, and CD’s derivatives [62,65,86,87]. In the PEI1.8-triMan conjugate, the carbohydrate fragment GlcNAc is in a linear form, while oligosaccharide alone is characterized mainly by a cyclic form of the reducing end; consequently, the graftization of the polymer by trimannoside was carried out according to the scheme (Figure 1b). In the NMR spectrum of PEI1.8-triMan, peaks appear in the region of 4.1–4.0 ppm and 5.0–4.8 ppm, and the peak of H5 shifts from 5.1 to 5.3 ppm (Figure 3a,b) according to the literature data on Man- and GlcNAc-containing sugars [88].

In the ^1^ H NMR spectra of HPCD-grafted conjugate (Figure 3c), both peaks of the initial components and new signals related to urethane bonds -N(H)-C(O) (formed as a result of the interaction of activated HPCD with spermine or PEI) were detected: the chemical shift -NH- at 7.05 ppm and the chemical shift -C(H)-O- at 2.8 ppm characterize the formation of urethane bonds [89,90]. Based on the integral intensity of the characteristic peaks of protons corresponding to triMan, PEI, and HPCD, the molar ratios of the components were calculated (Table 1) correlating with FTIR spectroscopy data.

#### 2.1.3. Physico-Chemical Properties of Mannosylated Ligands

The proposed conjugate’s structures and composition (Figure 1) were identified on the basis of conditions and sequence of synthesis reactions, the ratio of components, FTIR and NMR spectral data, and data on the number of mannosylated amino groups by spectrophotometric detection. Physico-chemical properties of synthesized conjugates are summarized in Table 1. Conjugates based on PEI10 have a molecular weight of 45–60 kDa, ligands based on PEI1.8–15–23 kDa, and a “light” 2 kDa amCD-triMan. An important parameter is the hydrodynamic size (Appendix A), which takes into account the hydrate shell of the ligand and the effective particle size. It is believed that the largest particles >200 nm settle well in the lungs and thus enter directly into the alveoli to macrophages through the use of aerosol form of drugs. Smaller particles can be injected intravenously since they do not cause a strong immune response, unlike high-mannose oligosaccharides or mannan. Therefore, a wide range of delivery systems can theoretically provide various options for the treatment of macrophage-associated diseases, considering the specifics of the disease. ζ-potential (Appendix A) of all the final conjugates (1, 2b, 3b) is in the range from −5 to +5 mV.

It is believed that positively or negatively charged (>10 mV) particles interact better with surface proteins or polysaccharides of cell membranes [91,92], which determines effective penetration. However, our strategy is aimed at avoiding non-specific capture of delivery systems, so the charge remains almost neutral (<5 mV). It is worth noting that the high positively or negatively charged (>15 mV) ligands could negatively affect the state of the biomembranes and blood components; for example, they may cause hemolysis of erythrocytes. Therefore, when using PEI as a carrier, it should be modified with neutral agents (such as CDs used in this work) that well-shield the charges groups and protect against aggregation at the same time. 

In addition to the size and charges for pharmaceutical preparations, the homogeneity of the obtained ligands is also important. The polydispersity indexes determined by DLS method (PDI, Table 1) of synthesized conjugates show quite acceptable characteristics in terms of homogeneity: it is approximately 0.3 for ligands with a PEI of 1.8 kDa and 0.5 for 10 kDa of polyethylenimine. The relative homogeneity of the samples can also be judged from the hydrodynamic size and the zeta-potential distributions of the particles (Appendix A), which are which are represented by a single peak.

Purification and homogeneity of samples were analyzed by HPLC. The chromatograms for HPCD-grafted PEI with triMan are shown in Appendix A. For ligands 2b and 3b, retention times are 4.5 and 3.1 min (dead time is 1.7 min). Polymers are eluted from the column by one main peak; however, the peaks are rather blurred to the right due to some polydispersity of the samples. Therefore, for further experiments, fractions were collected that correspond to the maximum peaks on the chromatograms ±0.5 min by retention time.

### 2.2. Conjugates’ Specificity to the Mannose Receptors on the ConA Model

One of the main parameters characterizing the effectiveness of drug delivery systems is affinity for targeted receptors. In this work, high affinity for mannose receptors (CD206) of alveolar macrophages is assumed. ConA was chosen as a model mannose receptor, for which relevance to CD206 was previously proven by methods of molecular dynamics and neural network analysis [43]. According to the literature, the CD206 domain shows a high similarity with mannan binding lectin A, which, in turn, is similar to ConA in terms of carbohydrate binding [93]. This is confirmed by numerous in vivo and in vitro studies [42]. Mannose-containing ligands, pre-selected by ConA assay, show high affinity (by flow cytometry) to CD206+ cells [84,94]. Similarly, mannose-containing ligands, pre-selected by ConA assay, are absorbed by macrophages via a receptor-mediated internalization mechanism [41] and are bio-distributed to lungs due to adsorption by macrophages [50]. All these data support the relevance of using ConA as a mannose receptor model. In this work, we used a previously validated technique [25] for determining the parameters of receptor-ligand binding using FTIR spectroscopy to these multicomponent systems.

Analytically significant absorption bands in the FTIR spectra of ConA are amide I and II (Figure 4a and Appendix A), which are two major bands of the protein infrared spectrum. The amide I band (between 1600 and 1700 cm^−1^) is mainly associated with the C=O stretching vibration (70–85%) and is directly related to the backbone conformation. Amide II results from the N-H bending vibration (40–60%) and from the C-N stretching vibration (15–40%). The ConA spectrum is also characterized by the amide A absorption band of N-H bonds 2800–3000 cm^−1^ (Figure 4a). At the same time, during complexation with a PEI-triMan ligands, the intensity of the characteristic bands of the protein changes (Figure 4a): the intensity of amide I decreases; in contrast to amide II, the minor peaks of amide III and IV increase and become more noticeable, which indicates the formation of hydrogen bonds, van der Waals, and electrostatic interactions between amino acid residues of the protein and carbohydrate ligand [25,42,43,44,45,95,96,97,98,99,100,101,102,103,104].

Recently, we showed using molecular dynamics that the “amide” atoms of ConA binding site are involved in complexation: Asn14 (N)–Man (O3), Asp16 (H)–Man (O5), and Arg222 (H)–Man (O2) [43]. This determines the observed changes in the FTIR spectra of the protein when binding to the mannosylated ligand (Figure 4a and Appendix A). At the same time, Arg228, Asn14, Asp208, and Leu99 interact with the terminal mannosyl residue through a network of hydrogen bonds, which explains the appearance of absorption bands in the region of 1380–1480 cm^−1^. Additionally, the ligand–protein complex is stabilized by stacking interactions (π-CH) of Tyr12 and Tyr100 with mannopyranoside planes, which is reflected in a change in the microenvironment of aromatic tyrosine systems: C–C skeletal deformations (1500 cm^−1^, overlapped with amide II), C–O stretching vibration of phenol (1270 cm^−1^), and in-plane C–H deformation vibration (1173 cm^−1^) [105].

The dissociation constants were calculated based on the dependence of the amide I and II peak intensity in the second-derivative FTIR spectra (Figure 4b). The complex formation process is described by the scheme ConA + nLigand ⇌ ConA·nLigand and can be linearized in Hill coordinates. The validity of the choice of intensity in the quality of the analytical signal is confirmed by the fact that the energy of valence and deformation vibrations also changes due to the formation of non-covalent protein-ligand bonds, which is demonstrated by short-wave shifts in the differential IR spectra of the second order (Figure 4c). This shift was additionally considered as an analytical signal, and thus, the degree of binding was calculated by several parameters at once (intensity of amides I and II and position of maxima and minima in the differential spectrum).

According to IR spectroscopic analysis, all the ligands considered interact with ConA in a ratio of 1:1 (within the margin of error) except for PEI1.8-triMan, which, being a compact molecule, can interact with a protein in a ratio of 2:1 (Table 2). PEI-triMan ligands are the most affine to mannose receptors among the synthesized ligands (Table 2): −lg *K*_d_ reaches a value of 6.5. PEI-triMan ligands grafted with HPCD are also highly affine. The lightest amCD-triMan ligand demonstrates an affinity for mannose receptors comparable to dimannosides [48], but its advantage is potentially higher loading capacity (LC) due to the presence of CD.

In our opinion, the most optimal and universal system is ligand 2b (HPCD-PEI1.8-triMan, Figure 1b); it has a relatively low molecular weight and therefore is more biodegradable than PEI10. However, at the same time, ligand 2b performs the function of active targeting due to its high affinity to mannose receptors and its LC up to 25% by weight of MF or another antibiotic with an adjuvant.

Earlier, using computer modeling [43], we showed that the binding of triMan and triMan-GlcNAc_2_ [46,48,95,96,99] with ConA occurs according to an “improved mechanism” in relation to single Man: an average of 15 hydrogen bonds are formed, and 5–9 are additionally mediated by water. In comparison with the mode of binding by proteins of single Man residues, the following amino acid residues are additionally involved: Asp16, Thr15, Pro13, Tyr12, Asp208, Thr226, Glu98, Arg228, and Gly224. This determines the strength of complexes with ConA: −lg *K*_d_ 5 for triMan and 6 for triMan-GlcNAc_2_. Since the oligosaccharide used in the work was attached by the reducing GlcNAc end (linker analog) to the polymer or CD, it is relevant to compare it with triMan. The affinity of the ligands 2a, 2b, and 3a is 0.5–1.5 orders of magnitude higher than that of the natural ligand trimannoside. Ligands 2a and 2b are comparable and even superior in affinity to mannan (−lg *K*_d_ 6).

### 2.3. Moxifloxacin Loading into Molecular Containers

As discussed above, the fourth-generation fluoroquinolone moxifloxacin was chosen as the main active component due to its wide spectrum of action and applicability for the treatment of respiratory tract diseases. As discussed above, even such strong antibiotics cannot cope with resistant or dormant bacteria. Targeted drug delivery and the use of adjuvant enhancers will help to combat them. In this section, the physicochemical parameters of MF loading into molecular containers are considered.

#### 2.3.1. FTIR Spectroscopy of MF and MF Conjugate

FTIR spectra of MF– and MF–ligand complexes are presented in Figure 5a and Appendix A. Spectra showed aromatic C=C stretching at 1621 (C=O and C=C), 1515, and 1448 cm^−1^; carboxylic acid C=O stretching at 1715 cm^−1^; C–N stretching at 1320–1380 cm^−1^; and C–F stretching at 1188 cm^−1^. The formation of non-covalent MF complexes with ligands is accompanied by a decrease in the intensity and a change in the shape of these peaks (Figure 5a and Appendix A). Confirmation of the change in the microenvironment of the quinolone structure of MF is the shift of the peak from 1448 to 1454 cm^−1^. An increase in the resolution of the spectrum can be achieved by differentiating and isolating thin components (Figure 5b). The observed shifts in the position of peaks in the differential spectrum to the high-frequency region correspond to changes in the vibrational energy of C=C, C=O, and C–N bonds in the composition of quinolone and pyrrolidine-piperazine structures following a change in the microenvironment when MF is included in the CD cavity or interaction with the polymer chain of conjugates. Based on the dependences of the intensity or position of the analytical peaks in the FTIR spectra of MF (Appendix A), the parameters of the interaction of MF with ligands were calculated (Table 3). The calculated values of constants in the order of magnitude are 5 × 10^−3^–10^−4^ M, which exceeds the previously described values for levofloxacin [25]. The constants for complexes MF–(mannosylated polymers) are comparable to those for the CD–MF complex [89,106,107,108]. At the same time, polymers grafted with CDs demonstrate values of the dissociation constants of complexes with MF of the order 5 × 10^−5^–5 × 10^−6^ M, while the drug capacity increases by 4–10 molecules for PEI1.8 and PEI10, respectively. Such high-capacity drug carriers can demonstrate the effects of improving pharmacokinetic parameters.

#### 2.3.2. NMR Spectroscopy of MF and MF Conjugate

^1^ H NMR spectra of MF in free form and in the form of non-covalent complexes with ligands 2a and 2b (PEI1.8-triMan and HPCD-PEI1.8-triMan) are presented in Figure 6. The interactions of HPCD and PEI as components of molecular containers with MF can be characterized by induced chemical shifts Δδ, equal to the difference of chemical shifts in the complex and single substances (Table 4) [89,109,110]. The interaction of MF with the PEI1.8-triMan polymer chain leads to induced shifts towards a strong field (Table 4), which corresponds to an increase in the hydrophobicity of the microenvironment of drug molecules, which indicates the successful encapsulation of MF into a molecular container. The inclusion of an apolar fragment of the MF molecule into the host hydrophobic cavity of HPCD (Figure 6c) induced a shielding of the inner protons of the glucose units of HPCD, namely H3 and H5, whereas the protons on the exterior of the torus (H1, H2, and H4) were relatively unaffected; this was previously shown in systems where CD formed inclusion complexes with Lev, EG [23], atropine, nitrobenzene, and nicardipine [111,112,113,114,115].

The inclusion of MF into the HPCD cavity is accompanied by induced chemical shifts of MF protons (Table 4), mainly upfield due to an increase in the hydrophobicity of the environment of the guest molecule. When MF interacts with molecular containers, the induced shift of the proton 49 is knocked out of the common vector, which is probably due to the presence of a neighboring electronegative atom F, which pulls the electron density more efficiently inside the CD cavity. The protons MF 30, 31, and 38 of the piperidine and cycloprapane rings (Figure 6a) are characterized by stronger shifts towards an upfield in the case of a polymer ligand grafted with HPCD in comparison with PEI1.8-triMan, and for the remaining protons, on the contrary, downfield shifts are observed due to the greater donor capacity of PEI nitrogen atoms than OH groups of HPCD. Based on ^1^H NMR spectra, MF is included in the cyclodextrin by the piperidine end and is buried to the carboxyl group, the location of which is advantageous outside the HPCD cavity and is additionally stabilized by the atoms of the HPCD outer shell [23]. Thus, the structure of the MF inclusion complex in HPCD is similar to those described earlier for levofloxacin [23,111], with the only difference that the piperazine end is completely immersed in the HPCD cavity and not by half, as in the case of piperidine in levofloxacin. A similar conclusion was obtained earlier using computer modeling of fluoroquinolone–cyclodextrin complexation processes [107,116].

### 2.4. Drugs Interaction with Bacteria

#### 2.4.1. Antibacterial Activity of MF against *B. subtilis*

The use of drug delivery systems as well as adjuvants to drugs allows increasing the therapeutic potential of the drug due to several aspects: (1) due to the formation of non-covalent complexes with a molecular container, the molecules of the therapeutic agent are more stable, and their active concentration is higher; (2) the ligand+adjuvant increases the permeability of the drug into cells; (3) adjuvant inhibits efflux proteins; (4) the delivery system allows one to load a larger amount of the drug and slowly release it while maintaining a given level agent concentrations; and (5) the rate of excretion from the body decreases. The data on item 4 were obtained and analyzed earlier by us: effective action time of levofloxacin (time of 80% release) was increased from 0.5 to 20–70 h due to mannosylated HPCD-PEI conjugates [25]. Point 5 is confirmed by the literature data: due to the conjugation of antibiotics with polymeric (polyelectrolyte) delivery systems, the half-life of fluoroquinolones and their time of effect are increased [5,117].

The role of adjuvants: The authors recently studied the effect of allylbenzenes and terpenoids on Gram-positive and Gram-negative bacteria [23,25]: both individual substances and synergists of antibiotics. It has been shown that among other allylbenzenes and terpenoids studied, EG, menthol and safrole most powerfully enhance the antibiotic. By themselves, these components of essential oils are not capable of efficiently defeating bacterial infection, but they have proven themselves as antibiotic enhancers (Figure 7).

The following is an experiment on points 1–3, which is a model of the action of a combined drug (MF + delivery system + adjuvant) in comparison with control and simple MF in the cell medium. The dependence of the number of surviving cells on the incubation time with MF, MF in the ligand, and MF enhanced with adjuvants EG, safrole, and menthol as well as with EG, safrole, and menthol themselves are presented in Figure 7 and Appendix A. By itself, the drug-free delivery system has practically no effect on cell growth (Figure 7). However, with loaded MF, positive effects of the conjugate on MF action are observed: amCD-triMan does not have a statistically significant effect on MF action up to 24 h; however, with prolonged observation >36–48 h, the effect of MF ends due to MF destruction [106]; on the contrary, MF in a complex form continues to be active. Similar results are described for polymer and 3D matrix structures of cyclodextrins in [104]. «Heavier» mannosylated PEI10 (ligand 3a, Appendix A) makes the antibiotic more active in comparison with the simple form at a concentration of 30 ng/mL, most probably due to the increase in the adsorption degree. The addition of adjuvants (0.1 mg/mL), especially EG and menthol, increases the number of defeated pathogen cells and MF activity by 20–30%, while EG, safrole, and menthol themselves have only a marginal effect on the cells’ viability. In addition, the effect of adjuvants is noticeable even with long observation > 24 h, when the active concentration of MF is insufficient to inhibit bacterial growth; however, in systems with adjuvants, MF continues to restrain bacterial reproduction. The effects of both molecular containers and adjuvants as antibiotic enhancers are vividly demonstrated (Figure 7). The triple formulation MF+ligand+adjuvant proved to be a particularly powerful antibacterial agent.

To clarify the mechanisms of action (membrane permeability and efflux inhibition) of the components, CLSM and fluorescent studies of the interaction of drugs with bacteria were carried out.

#### 2.4.2. Confocal Laser Scanning Microscopy (CLSM)

This paragraph shows the role of EG adjuvant and molecular containers and the mechanism of their action together and separately. Namely, we studied the effect of ligand and adjuvant on membrane permeability for an antibiotic and reflux inhibitor; therefore, the purpose of the paragraph is to visualize these processes using CLSM. Dox was taken as the model antibiotic as a widely used drug and fluorescent dye in the visible region, which is necessary for registration on a confocal microscope with laser excitation. 

We studied six samples:(1)Control Dox to study the penetration and accumulation of free matter in cells;(2)Dox in a polymer ligand to study the effect of polymer on the adsorption efficiency on the cells and penetration of Dox;(3)Dox in a polymer ligand with grafted CD tori to clarify the role of CD tori in the creation of defects in the membrane and increased Dox (LC) capacity;(4)Dox with EG as an agent enhancing the membrane permeability and efflux inhibitor;(5)Dox with preincubated EG only as efflux inhibitor so as to exclude the effect of adjuvant on the membrane permeability due to the desorption of EG;(6)Combined formulation Dox + ligand with grafted CD tori + EG.

Confocal images of the interaction of Dox (red) and FITC-labeled ligands (green) with *B. subtilis* are shown in Figure 8 (full version—Appendix A).

*Free Dox* accumulates weakly in cells (Figure 8a, Appendix A). The maximum concentration among the studied times is reached after 15 min. Over time, Dox is released from the cell by pumping proteins in a process of efflux. Therefore, the drug practically does not accumulate in the cells during the observation period of 90 min. 

*The effect of adjuvant EG on the accumulation of Dox.* When an adjuvant is included in the system simultaneously with Dox, an increase in the degree of Dox accumulation in bacterial cells is observed (Figure 8b,c, Appendix A) due to the occurrence of defects in the cell membrane [17] and inhibition of pump proteins [10,11]. The effect of gradual perforation of the cell membrane was confirmed by a decrease in CFU over time during incubation of cells with EG (Appendix A). When EG acts in two directions (creates defects in the membrane and inhibits pump proteins), the effect of enhancing drug penetration increases over time (Figure 8b, Appendix A). However, at the same time, during the observation, EG did not have enough time to block the efflux of proteins because the effluxed Dox was observed in solution.

*Efflux inhibition by EG without its effect on the membrane permeability.* EG preincubated with cells for an hour and then removed from the solution and desorbed from the membrane (by cells washing with 0.01% SDS, a control turbidimetry and FTIR experiments showing that SDS in 0.01% has no effect on the membrane; results are presented in Appendix A and in Methods section) can act primarily as an inhibitor of efflux proteins and, a to lesser extent, destroy the membrane. The maximum effect of EG inside the cells and washed outside is reached after 30 min (Figure 8c, Appendix A). EG effectively “turned off” the pump proteins and a significant amount of Dox was able to penetrate into the cells. Visually, the presence of Dox outside the cells is noticeably less than in the previous case, which means that the efflux efficiency is achieved in a few tens of minutes, while the EG starts to increase the membrane permeability immediately (Figure 8b), which is consistent with the change of A600 from incubation time (Appendix A).

*The effect of molecular containers (ligands) on the accumulation of Dox.* As discussed above, the use of a molecular container is expected to increase the degree of adsorption and concentration of the drug in the vicinity of the cells surface, penetration, and stabilization and maintain the concentration level above the threshold for a long time (at least a few days). According to experimental data, due to the adsorption of polymer particles on the cell membrane, the accumulation of Dox increases with a time of 0–15–30–90 min (Figure 8d,e, Appendix A). The ligand effect is especially noticeable when comparing micrographs of free Dox accumulation (Figure 8a,d). 

The greatest effect is achieved for the polymer ligand 2b grafted with HPCD tori since Dox is more efficiently loaded into the hydrophobic cavities of the CD, as earlier reported for similar conjugates [25,86,106]. Furthermore, CD additionally enhances penetration through the membrane [52].

The *additive effect* is observed in the system (Dox + EG + FITC-HPCD-PEI1.8-triMan), in which the penetration of Dox increases due to the addition of the above effects, while the greatest contribution is made by efflux inhibitor EG. Moreover, the ligand loads both EG and Dox, and since the distribution of components is uniform for such double-inclusion complexes (earlier shown for levofloxacin and EG [25]), both EG and Dox molecules are likely to be next to each other at once. The first one will create a defect in the membrane where the target substance, Dox, will penetrate directly through the hole formed nearby. Thus, the complex formulation antibiotic+adjuvant in a drug delivery system is an effective way to accumulate the drug in the target bacteria.

#### 2.4.3. Fluorescent Determination of MF and EG Adsorbed and Absorbed by Cells

CLSM allows semi-quantitative observation of the studied processes of increased penetration due to defects in the membrane and inhibition of efflux. Fluorescence spectroscopy was used to quantify the effects of EG. The essence of the experiment was to incubate cells with MF or MF+EG during 14 h and then study the distribution of substances inside/on the cell surface or in solution depending on the time and correlate the results with the visualization obtained by CLSM.

Table 5 shows the concentrations of MF and EG depending on the incubation time of substances with *B. subtilis*. The concentration of MF in the solution decreases due to adsorption on the surface of the bacterial cell wall and penetration of fluoroquinolone into the cells. At the same time, the concentration of absorbed MF did not exceed 2.8 μM and decreased due to efflux, which was observed also in a confocal microscope.

However, EG accelerates the penetration of MF due to increased membrane permeability, and in 15 min, the maximum concentration of the antibiotic is reached inside the cells (4.4 μM). EG does not affect maximum time statistically significantly. Thus, EG allowed the accumulation of MF by 60% more.

Pump proteins work in cells, throwing MF and EG out. Approximately 0.5% of the total concentration of EG is absorbed, and the remaining molecules are adsorbed on the membrane. EG absorbed inside the cells inhibits pump proteins; however, this does not happen instantly but after tens of minutes, as shown using CLSM. After inhibition of efflux, EG allows maintaining twice the concentration of MF inside the cells for several hours. After prolonged incubation due to significant damage to the cell membrane and partial lysis, substances leak out (Appendix A). A significant part of the cells in the EG system was destroyed, while free MF could not completely inhibit the growth of the number of bacteria. Thus, a way was found to increase the effectiveness of antibiotics by several times through the use of polymer ligands and adjuvants. The obtained data are consistent with CLSM data, which link the mechanism of penetration and efflux of Dox and MF.

### 2.5. Drug Delivery System Interactions with Macrophages

The key experiment demonstrating the therapeutic potential of the developed systems is the study of the interaction of delivery systems with macrophages. Seven days after adding PMA, we derived macrophage-like cells from THP-1 cell line as described before [88]. Most of these cells were CD206-positive according to immunocytochemical evaluation (Figure 9). We also determined that CD206-positive cells effectively phagocytosed the FITC-labeled polymer as shown in Figure 9. Cytometry assay determined that 95.5% macrophage-like cells were FITC-positive after adding HPCD-PEI1.8-triMan, and 61.7% were FITC-positive after adding HPCD-PEI1.8-Man (Figure 10). Primary culture of human dermal fibroblasts (HDF) was used as a negative control for CD206 receptors. Appendix A show immunocytochemical evaluation of CD206, phagocytosis assay, and flow cytometry data on FITC+ distribution of fibroblasts. Fibroblasts express CD206 in small amounts and are able to non-specifically phagocytize mannosylated particles. Cytometry assay determined that 17.4% of human dermal fibroblasts (CD206-negative) cells were FITC-positive after adding HPCD-PEI1.8-triMan, and 4.3% were FITC-positive after adding HPCD-PEI1.8-Man (Appendix A). The phagocytic activity of fibroblasts against polymer mannosylated particles is low, which confirms the specificity of CD206 macrophage delivery systems to mannose recipes. Thus, the developed drug carriers fulfilled three standards: high macrophage uptake, high drug-carrying capacity, and prolonged and enhanced antibacterial effect with adjuvants, which determines the potential applicability for the use of complex combined drugs for medical purposes.

## 3. Materials and Methods

### 3.1. Reagents

Carbonyldiimidazole–CDI was obtained from GL Biochem (Shanghai, China) Ltd. via an intermediary Himprocess (Moscow, Russia). ConA was purchased from Paneco-ltd (Moscow, Russia). D-mannose, PEI 1.8 kDa and PEI 10 kDa (branched); and fluorescein isothiocyanate (FITC), doxorubicin, DMF, DMSO, Et_3_N, 2-hydroxypropyl-β-cyclodextrin (HPCD), 1M 2,4,6-trinitrobenzenesulfonic acid, and moxifloxacin hydrochloride were obtained from Sigma Aldrich (St. Louis, MI, USA). Mono-(6-(1,6-hexamethylenediamine)-6-deoxy)-β-cyclodextrin (amCD) was supplied by Dayang Chem (Hangzhou, China) Co., Ltd. Mannotriose-di-(*N*-acetyl-D-glucosamine) was obtained from Dayang Chem (Hangzhou) Co., Ltd. Other chemicals, namely NaBH_4_, salts, and acids, were from Reakhim Production (Moscow, Russia). All chemicals were of analytical grade or higher and were used without additional purification.

### 3.2. Synthesis and Purification of Mannosylated Conjugates

Activated HPCD was synthesized as described earlier [25]. Briefly, the slow drop-by-drop addition of HPCD solution in water 1 mL (0.4 mM) to the 10 mL CDI solution in DMSO (1.5 mM) led to the formation of the HPCD–CDI_7_ composition at the output. 

Ligands **1** (Figure 1a), **2a**, and **3a** (Figure 1b): Three specimens containing (1) 180 mg of PEI1.8 and 580 mg of triMan-GlcNAc_2_, (2) 100 mg of PEI10 and 500 mg of triMan-GlcNAc_2_, and (3) 150 mg of amCD and 250 mg of triMan-GlcNAc_2_ were dissolved in 5 mL 0.01, 0.1, and 0.001 M HCl, respectively, followed by immediate alkalinization (1M NaOH) to pH 9–10 and incubation at 50 °C for 4 h (here and further, the Biosan ES-20/60, 160 rpm shaker incubator was used) and subsequent reduction of Shiff bases. Samples of solutions from systems 2 and 3 were taken: conjugates **2a** and **3a**. Further crosslinking of PEI amino groups in systems 2 and 3 with activated HPCD was carried out by addition of 7- and 12-fold molar excess of HPCD–CDI_7_ relative to PEI chains, respectively, followed by incubation at the same temperature for an hour. Samples **2b** and **3b** were obtained. Conjugates purification was carried out by dialysis at 22 °C against water in three stages of 12 h with water replacement: ligands **2a**, **2b**—cut-off 3 kDa; ligands **3a**, **3b**—cut-off 6–8 kDa; ligand **1**—cut-off 2 kDa. All samples were freeze-dried for two days at −60 °C (Edwards 5, BOC Edwards, UK). The purification of the samples was performed by HPLC gel filtration in a Knauer chromatography system (Knauer, Berlin, Germany) on Diasfer-110-C18 column (BioChemMack, Moscow, Russia): grains—6 μm, size 4 × 150 mm. The eluent was CH_3_CN-H_2_O (90:10, v:v); the elution rate was 0.8 mL/min, 25 °C. Due to high polydispersity of the conjugates, fractions corresponding to maxima ± 0.5 min by retention time were used in the experiments. The chromatogram of the resulting conjugates is shown in Appendix A. The degree of mannosylation was calculated according to spectrophotometric titration of amino groups (before and after mannosylation) with 2,4,6-trinitrobenzenesulfonic acid [25,101]. Mainly, primary amino groups were detected.

### 3.3. FTIR Spectroscopy

ATR-FTIR spectra of samples’ solutions were recorded using a Bruker Tensor 27 spectrometer equipped with a liquid-nitrogen-cooled MCT (mercury cadmium telluride) detector. Samples were placed in a thermostatic cell BioATR-II with ZnSe ATR element (Bruker, Germany). The IR spectrometer was purged with a constant flow of dry air (Jun-Air, USA). FTIR spectra were acquired from 900 to 3000 cm^−1^ with 1 cm^−1^ spectral resolution. For each spectrum, 50–70 scans were accumulated at 20 kHz scanning speed and averaged. Spectral data were processed using the Bruker software system Opus 8.2.28 (Bruker, Germany), which includes linear blank subtraction, baseline correction, differentiation (second order, 9 smoothing points), min–max normalization, and atmosphere compensation [25,118]. If necessary, 5-point Savitsky–Golay smoothing was used to remove noise. Peaks were identified by standard Bruker picking-peak procedure.

### 3.4. NMR Spectroscopy

Next, 12 mg of the sample was dissolved in 700 μL of D_2_O. ^1^H NMR spectra of the solutions were recorded on a Bruker Avance 400 spectrometer (Germany) with an operating frequency of 400 MHz. Chemical shifts are given in ppm on the δ scale relative to hexamethyldisiloxane as an internal standard. The analysis and processing of NMR spectra were performed in the program MestReNova v. 12.0.0–20080 (Mestrelab Research S.L.).

### 3.5. Dynamic Light Scattering (DLS)

The particle sizes and zeta potentials were measured using a Zetasizer Nano S «Malvern» (England) (4mW He–Ne laser, 633 nm, scattering angle 173°) in 0.01M PBS (pH 7.4). The experiment was performed three times in a temperature-controlled cell at 25 °C. Autocorrelation functions of intensity fluctuations of light scattering were obtained using the correlation of the Correlator system K7032-09 «Malvern» (Worcestershire, UK). Polydispersity indices were calculated based on particle size distributions (Table 1). Experimental data were processed using «Zetasizer Software» (v. 8.02).

### 3.6. Mathematical Calculations and Equations

(1)Calculation of the dissociation constants ConA ligand. Consider the equilibrium: ConA + n ligand ↔ ConA nligand, where *K*_d_ = [ligand]^n^ · [ConA]/[ConA nligand].(2)First, fitting of the curves of change of the analytical signal ξ (peak intensity or peak position in FTIR native or second-derivative spectra) versus concentration of the ligand was carried out using the Hill equations: (1) ξ = ξ_∞_ · [ligand]^n^/([ligand]^n^ + *K*^n^), where ξ_∞_—horizontal asymptote; (2) ξ = ξ_0_ + (ξ_∞_ − ξ_0_) · [ligand]^n^/([ligand]^n^ + *K*^n^). Second, calculation of n and *K*_d_ values was performed by Hill’s linearization in n-binding site model: lg (θ/(1 − θ)) = n · lg [L] − lg *K*_d_, where θ = | (ξ − ξ_0_)/(ξ_∞_ − ξ_0_) | is a fraction of the bound substance. n in Hill equation—number of ligand molecules per one ConA subunit.(3)Calculation of the dissociation constants drug ligand. Consider the equilibrium: drug + ligand ↔ drug ligand, where *K*_d_ = [ligand] [drug]/[drug ligand]. The calculations were performed similarly to the above point with the difference in calculating the number of drug molecules (MF) per conjugate (ligand) molecule. The number of MF molecules retained by one conjugate molecule in CD tori (N_CD_) is assumed to be equal to 90% of the total number of CD tori. The number of MF molecules held by one conjugate molecule by a polymer matrix (N_polymer_) is calculated based on the drug: ligand ratio, when the degree of binding θ is 95%, minus the drug molecules in the CD tori.(4)Calculations for the following items are performed when C_ligand_total_ = 0.0001 M, C_MF_total_ = 0.001 M.(5)Entrapment efficiency of MF into conjugates [18,24,119]: EE (%) = 100 · (MF amount in form of complex with ligand)/(total MF amount). *K*_d_ = N · (C_MF_total_ − C_MF_bounded_) · (C_ligand_total_ − C_MF_bounded_/N)/C_MF_bounded_. From the last equation, find C_MF_bounded_, and then, EE = C_MF_bounded_/C_MF_total_.(6)Loading capacity of drug MF into conjugates [18,24,119]: LC (%) = 100 · (mass of loaded MF)/(mass of sample) = 100 · (number of drug molecules per ligand molecule) · (molar mass of MF)/(molar mass of ligand). The number of drug molecules per ligand molecule is calculated in paragraph 2.(7)Statistical analysis of obtained data was carried out using the Student’s *t*-test Origin 2022 software (OriginLab Corporation). Values are presented as the mean SD of three experiments.

### 3.7. Confocal Laser Scanning Microscopy

*B. subtilis* cells (NCIB 8054, overnight culture, 10^7^ CFU) were centrifuged twice (Eppendorf centrifuge 5415C, 10 min, 12,000× *g*) and washed with 0.01M PBS from the culture medium. Next, the cells were incubated for 15–30–90 min at 37 °C with 500 µL of solutions indicated in the caption to Figure 8 and Appendix A. A special case was a pre-incubation of EG with cells for 1 h and washing with 0.01% SDS and a two-time washing-centrifugation of PBS and subsequent incubation with MF 15-30-90 min to study the efflux due to EG penetrating into the cells. The cells were centrifuged twice with PBS washing. The cell centrifuge was suspended in 200 µL of PBS, followed by the addition 100 µL of a 16% agarose solution at 45 °C to solidify the cell suspension in the wells of a fluorescent plate.

Fluorescence images were obtained by confocal laser scanning microscope (CLSM) Olympus FluoView FV1000 equipped with both spectral version scan unit with emission detectors and transmitted light detector. CLSM is based on a motorized inverted microscope Olympus IX82. The excitation wavelength 488 nm (multiline Argon laser) and dry objective lens Olympus UPLSAPO 40X NA 0.90 were used for the measurements. Laser power, sampling speed, and averaging were the same for all image acquisitions. The scan area was 80 × 80 μm^2^. FITC and Dox fluorescence were collected using the emission windows set at 510–540 nm and at 590–680 nm, respectively (emission fluorescence spectra are presented on Appendix A). The signals were adjusted to the linear range of the detectors. Olympus FV10 ASW 1.7 software was used for acquisition of the images. FITC fluorescence is shown in green, Dox is red, and the image on the light is gray.

### 3.8. Determination of MF and EG Concentrations Absorbed and Adsorbed by Cells

*B. subtilis* cells (NCIB 8054, overnight culture, 10^6^ CFU) were centrifuged three times (Eppendorf centrifuge 5415C, 10 min, 12,000× *g*) and washed with 0.01M PBS from the culture medium. Next, the cells were incubated for 15–45–120 min and 14 h at 37 °C with 200 µL of solutions indicated in Table 4. The cells were centrifuged (10 min, 12,000× *g*) and washed with PBS. The supernatant was used to determine MF and EG not bound to cells (C in solution). The cell centrifuge was suspended in 200 µL of PBS containing 0.02–0.05% SDS, followed by incubation at 37 °C 30 min. By centrifugation (10 min, 12,000× *g*), 2 fractions were obtained: a supernatant containing EG and MF corresponding to adsorbed substances (C adsorbed) and a cellular centrifugate washed from non-absorbed substances. The second fraction, resuspended in 1000 µL of PBS, was exposed to an ultrasonic probe for 30 min. Then, the amount of absorbed substances was determined in the cell lysate (C absorbed). Cells were also incubated with ligands without MF and EG, and these data were used as a control and background. Fluorescence of MF and EG was measured using a Varian Cary Eclipse spectrofluorometer (Agilent Technologies, Santa Clara, CA, USA) at 20 °C: λ_exc_ = 280 nm, λ_emi_(EG) = 320 nm, λ_emi_(MF) = 465 nm, slit 10/10 nm. 


*Control experiment on the effect of SDS on cell membrane permeability and drug absorption.*


First, 2 mL of cell suspension (5 × 10^6^ CFU/mL) was added to 2 cuvettes, and then, 20 mL of 10% SDS was added to the first one. Optical absorption at 600 nm as a correlate of CFU and membrane integrity was measured for two hours. Relative to the control solution, the CFU change did not exceed 10% of the initial value (Appendix A).FTIR spectroscopy data. The experiment consisted of incubating cells with Dox with and without the addition of SDS, then separating the supernatant and cells and determining the integrity of the membrane; then, Dox content was determined. The observed parameter is the intensity and positions of the characteristic peaks of the aromatic system of doxorubicin. In the case of a significant effect of SDS on membrane permeability, changes in peak intensity would be observed in the IR spectrum.

Technique. The cells were washed from the medium as described above, then suspended in 500 µL PBS (5 × 10^7^ CFU/mL). Next, the cell suspension was incubated at 37 °C for two hours with 0.1 mg/mL of Dox-MCD (1:2) with and without addition of SDS to the final concentration 0.1%. By centrifugation (10 min, 12,000× *g*) 2 fractions were obtained with subsequent registration of FTIR spectra: a cellular centrifugate suspended in 50 µL PBS (Appendix A) and the supernatant. The intensity of the characteristic peaks of doxorubicin (C–C_arom_ 1400–1550 cm^−1^) practically do not depend on the addition of SDS both inside the cells (Appendix A) as well as in the supernatant. 

On the other hand, the FTIR spectra of bacteria (Appendix A) contain information about the structural organization of cells [88,119]. Lipids and lipoproteins are the contribution of valence vibrations of CH_3_–, CH_2_– groups (2820–2980 cm^−1^), valence oscillations C=O (amide 1), and deformation N-H (amide II)—for phospholipids 1240 cm^−1^ PO_2_^–^. According to the literature data [119], these peaks are the most sensitive to lipid membrane state, which was proven by cells incubation with surfactants. Since the characteristic peaks of the lipid membrane, surface proteins, and glycans practically do not change from the addition of 0.1% SDS, it can be assumed that there was no violation of the integrity of the membrane. Otherwise, amides I, II, and III (very sensitive to membrane state, 1450–1700 cm^−1^) would have decreased [119] due to the greater availability of IR radiation of internal proteins and DNA. Therefore, the concentration of SDS studied with a 2–10-fold additional excess (0.1%) for 1–2 h does not have a significant effect on the amount of the drug absorbed by the cells.

### 3.9. Antibacterial Activity of MF in Free, Complex, and Enhanced with Adjuvants Forms

The strain used in this study was *Bacillus subtilis* (NCIB 8054) from the National Resource Center All-Russian collection of industrial microorganisms SIC “Kurchatov Institute”). The culture was cultivated for 18–20 h at 37 °C to CFU ≈ 1 × 10^7^ (determined by A_600_) in liquid nutrient medium Luria–Bertani (pH 7.2) with stirring at 160 rpm. The study of the antibacterial effect of MF in liquid media was carried out using an overnight bacterial culture. The cell suspension was diluted to CFU ≈ 3 × 10^6^−5 × 10^6^ with LB liquid medium (pH 7.2), and 400 μL of sample (indicated in Figure 7 and Appendix A) was added to 4 mL of culture. The resulting systems were incubated at a temperature of 32 °C and stirred at 160 rpm in an ES-20 BIOSAN incubator shaker (Latvia) for 2–4 days. Aliquots were selected at certain intervals to measure absorption at 600 nm to determine the number of viable cells.

### 3.10. Experiments with Macrophages

#### 3.10.1. Cell Lines and Modelling

For the macrophages CD206+ phagocytose assay, a human monocyte cell line THP-1 (or human dermal fibroblasts (HDF) used as a negative control for CD206-) was used. Cells were obtained from the bank of cell lines of Lomonosov Moscow State University. THP-1 cells were cultured on RPMI-1640 (Gibco) supplemented with GlutaMAX™ Supplement (Gibco) buffered with 10 mM HEPES pH 7.4 containing 10% heat-inactivated FBS (Gibco) and 1% antimycotic antibiotic (HyClone) at 37°C and 5% CO_2_. Macrophage-like cells were derived by adding to THP-1 (0.5 million cells/mL) and 100 nM phorbol 12-myristate 13-acetate (PMA, p8139, Sigma) for 72 h. After 72 h, the medium was replaced, and cells were cultured for another 96 h.

#### 3.10.2. Phagocytosis Assay

Dry samples of HPCD-PEI1.8-triMan and HPCD-PEI1.8-Man conjugated with a FITC fluorescent label were diluted in PBS (PanEco, Moscow, Russia) with 10% DMSO. Macrophage-like cells were washed with Hanks’ solution, and HPCD-PEI1.8-triMan and HPCD-PEI1.8-Man were added at final concentration of 0.10 mg/mL and 0.089 mg/mL, respectively, in serum-free media. As a control, we used macrophage-like cells without adding polymer. For cytometry assay, cells were incubated for 40 min, washed trice with Hank’s solution, and scraped in Versen’s solution. Cell suspension was centrifuged at 300× *g* for 5 min at +4 °C, and the cell pellet was resuspended in a PBS solution. For immunocytochemistry assay, cells were incubated for 4 min, washed with PBS, and fixed with 4% formaldehyde solutions.

#### 3.10.3. Cytometry

Cytometry was performed on a BD FACSAria™ III Cell Sorter instrument, the fluorescence of the FITC-conjugated polymer captured by the cells was analyzed, the change in the percentage of cells fluorescing in the FITC channel was determined, and cells not incubated in the sample solution were evaluated (control).

#### 3.10.4. Immunocytochemistry

To block nonspecific binding sites, cells were incubated with a 10% solution of normal goat serum in PBS with the 1% bovine serum albumin BSA for 1 h at RT. Then, the samples were incubated with a solution of anti-CD206 antibodies (ab64693, Abcam, 1:100) or rabbit polyclonal control IgG (910801, Biolegend) as a control for 2 h at RT and subsequently with goat-anti-rabbit antibody conjugated with Alexa594 (A11037, Invitrogen, 1:1000). The nuclei were labeled with DAPI (Sigma-Aldrich, St. Louis, MO, USA). Samples were analyzed with a Leica DM6000B fluorescent microscope equipped with a Leica DFC 360FX camera (Leica Microsystems GmbH, Wetzlar, Germany).

## 4. Conclusions

Currently, a promising direction in the treatment of complicated infectious diseases of the respiratory tract (including resistant tuberculosis) is the development of complex combined forms of drugs based on fluoroquinolones. In this paper, two focuses on solving the problem of bacterial resistance are considered: (i) the addressed delivery of fluoroquinolones to macrophages by targeting mannose receptors CD206 and (ii) the selective effect on bacteria (and in the future on cancer cells) due to adjuvant-increased penetration of drugs and inhibition of efflux and, as a consequence, accumulation of drugs in the area of pathogens localization. Firstly, it has been shown that oligo- and polymer conjugates, i.e., mannan grafted with cyclodextrins and mannosylated PEI with grafted CDs, are high-capacity molecular containers capable of loading 8–20% by weight of the therapeutic agent and delivering it to macrophage mannose receptors, which is shown on the model protein ConA and cells. Flow cytometry has shown that annotated polymers are actively absorbed by macrophages: with linear mannose, 62% of cells phagocytized a ligand, and a high-affinity trimannoside conjugate absorbed 95.5% of macrophage cells. Secondly, CLSM was used to study the mechanism of action of an adjuvant as an agent that increases the permeability of the membrane and an efflux inhibitor together and individually as well as a molecular container. Efflux management is important for overcoming multidrug resistance. Therefore, the conducted research opens up new opportunities to influence complex macrophage-associated infections. The data obtained in combination with previous studies form the basis for choosing the optimal delivery system and adjuvant for a specific therapy task and possibly a specific patient.

## Figures and Tables

**Figure 1 pharmaceuticals-15-01172-f001:**
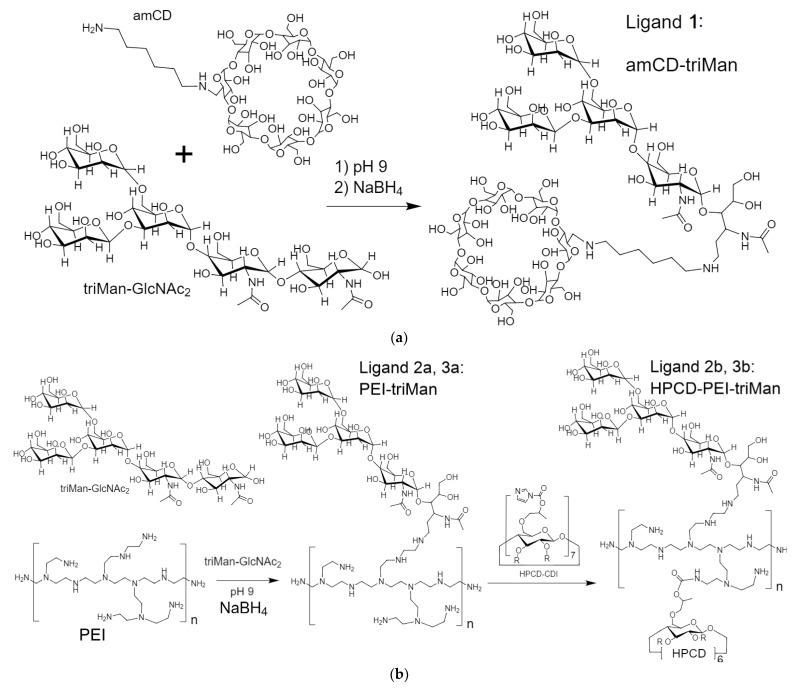
Scheme of synthesis of mannosylated conjugates: (**a**) amCD-triMan; (**b**) PEI1.8-triMan or PEI10-triMan and HPCD-PEI1.8-triMan or HPCD-PEI10-triMan. For PEI 1.8 kDa, n is in the range 3–5; for PEI 10 kDa, n is in the range 20–25.

**Figure 2 pharmaceuticals-15-01172-f002:**
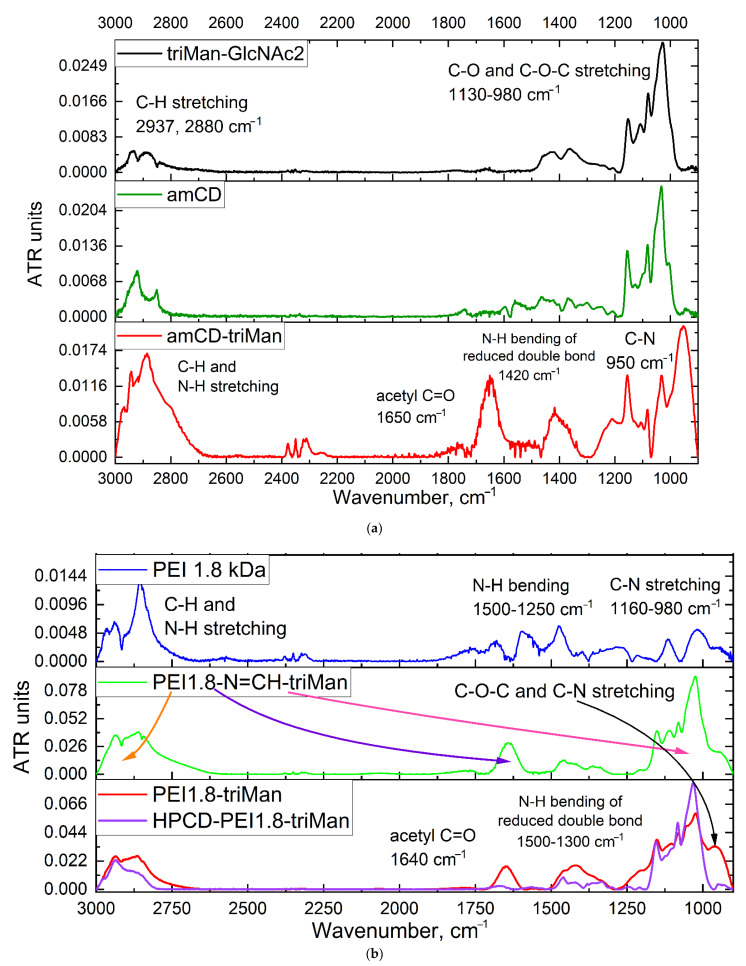
Fourier transform infrared spectra of: (**a**) triMan-GlcNAc_2_, amCD, amCD-triMan; (**b**) triMan-GlcNAc_2_, PEI 1.8 kDa, PEI1.8-N=CH-triMan, PEI1.8-triMan, and HPCD-PEI1.8-triMan. Aqueous solutions. T = 22 °C.

**Figure 3 pharmaceuticals-15-01172-f003:**
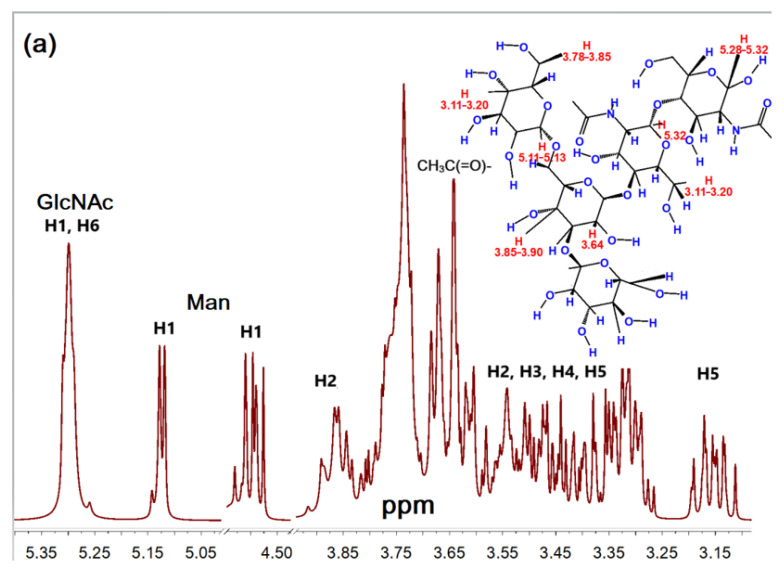
^1^H NMR spectra of (**a**) triMan-GlcNAc_2_, (**b**) PEI1.8-triMan, and (**c**) HPCD-PEI1.8-triMan with proton assignment. D_2_O, 400 MHz.

**Figure 4 pharmaceuticals-15-01172-f004:**
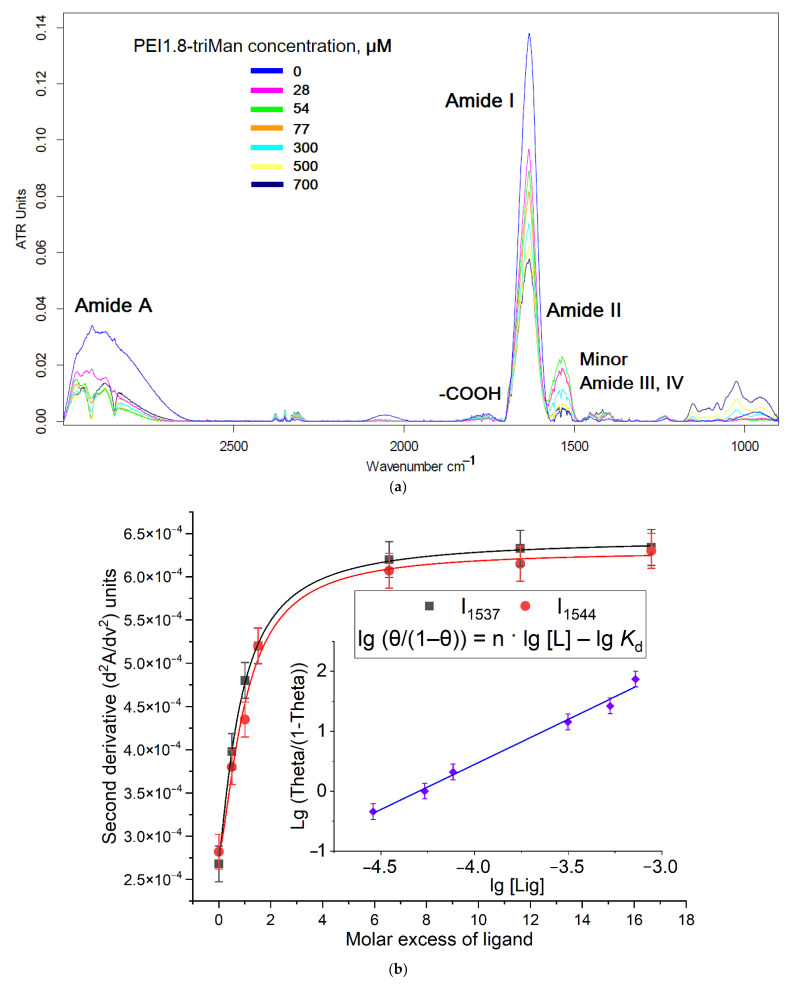
(**a**) Fourier transform infrared spectra of ConA and ConA complexes with PEI1.8-triMan. (**b**) Dependences of the intensity of amide I and amide II peaks in the FTIR spectra from (**a**). On the insert, the Hill linearization plot of the presented curves is given. (**c**) Second derivative (d^2^A/dν^2^) of FTIR spectra of ConA and ConA complexes with PEI10-triMan. The arrows show the shifts during the complexation of the protein with the increasing concentration of ligand. [L], the equilibrium concentration of the ligand; θ, proportion of bound receptor. C_0_(ConA subunit) = 0.1 mM. Natrium-phosphate buffer solution (0.02 M, pH 6.4). C(Ca^2+^) = C(Mn^2+^) = 1 mM. T = 25 °C.

**Figure 5 pharmaceuticals-15-01172-f005:**
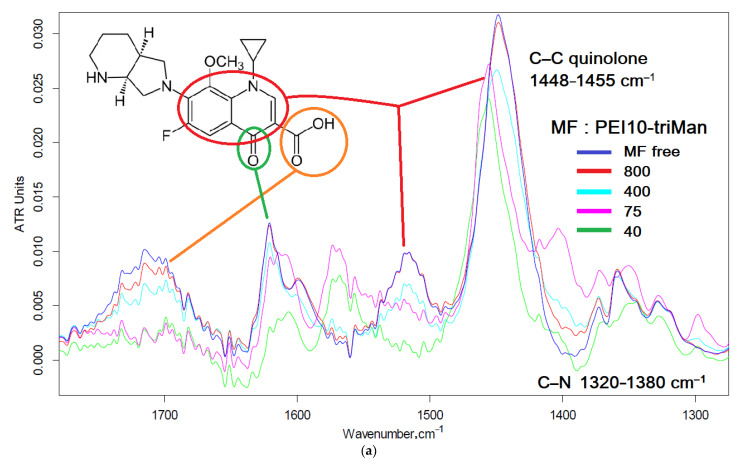
(**a**) FTIR spectra and (**b**) its second derivative (d^2^A/dν^2^) of MF and MF complexes with PEI10-triMan. Concentrations in (**b**), such as on A. 0.01 M HCl. T = 22 °C.

**Figure 6 pharmaceuticals-15-01172-f006:**
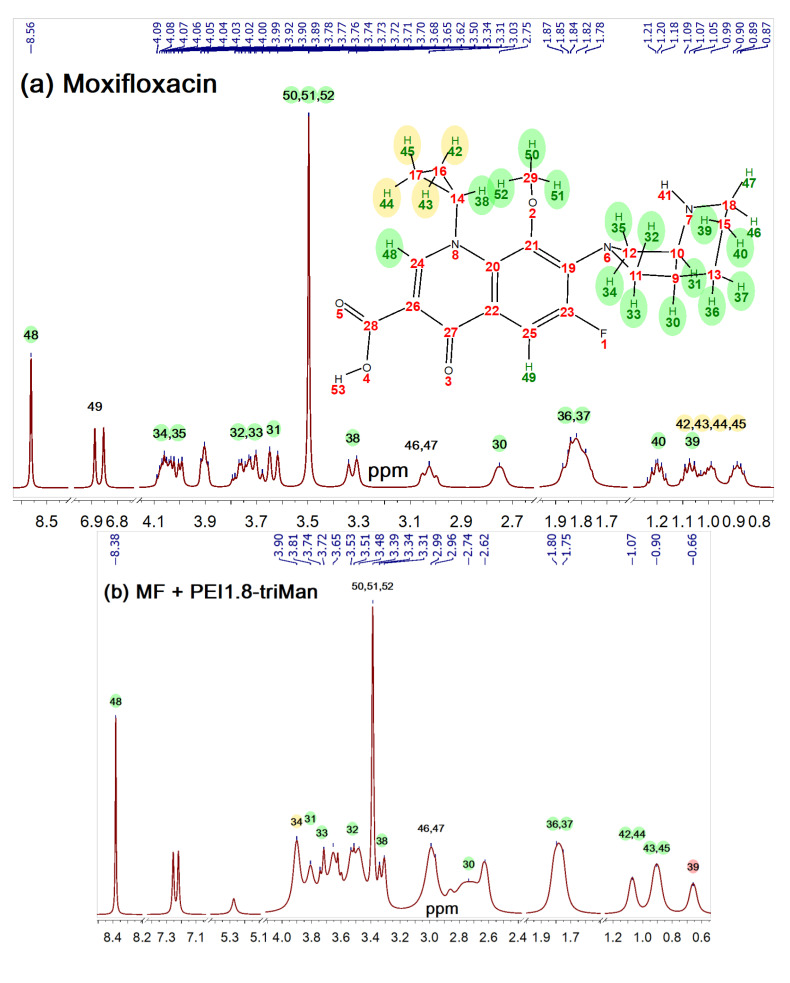
^1^H NMR spectra of (**a**) MF, (**b**) MF complexed with PEI1.8-triMan (w:w = 1:1), and (**c**) MF complexed with HPCD-PEI1.8-triMan (w:w = 1:1) with proton assignment. D_2_O, 400 MHz.

**Figure 7 pharmaceuticals-15-01172-f007:**
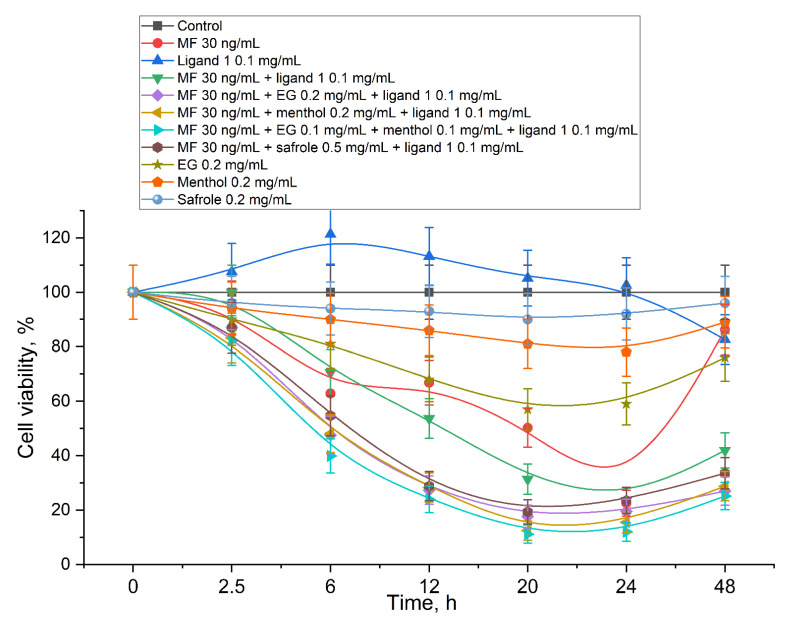
Cell viability of *B. subtilis* as a function of incubation time with drug samples. Major active component is MF; drug loading system—ligand 1 (amCD-triMan); adjuvants—EG, menthol, and safrole. CFU (0 h) = 3 × 10^6^. pH 7.4 (0.01 M PBS), 32 °C. The results are presented as the average of three parallel CFU measurements ± SD.

**Figure 8 pharmaceuticals-15-01172-f008:**
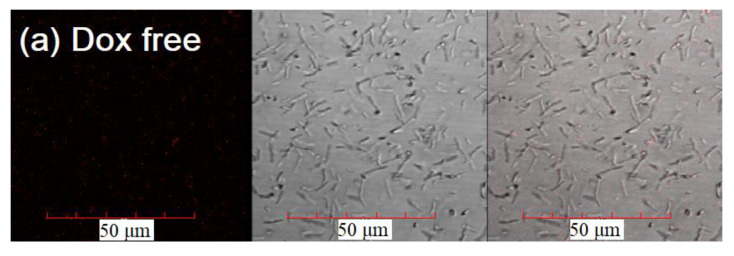
Confocal laser scanning images of *B. subtilis* with adsorbed and absorbed Dox and FITC-labeled ligands based on PEI, HPCD, and triMan. (**a**) Dox-free. (**b**) Simultaneously added Dox + EG (0.1 mg/mL) as a membrane destroyer and protein efflux inhibitor. (**c**) Dox + EG (0.1 mg/mL) only as a protein efflux inhibitor, where EG is added in advance and then washed using SDS. (**d**) Dox + FITC-HPCD-PEI1.8-triMan. (**e**) Dox + EG + FITC-HPCD-PEI1.8-triMan. The scale segment is 50 μm (division value is 10 μm); 30 min of incubation (except for point (**b**)). C_0_(Dox) = 1.3 μg/mL. C_0_(FITC) = 0.2 μg/mL; 3–4 channels are shown: red, Dox; green, FITC; gray, transmission light mode; and overlay. Emission fluorescence spectra of FITC and Dox obtained by CLSM are presented in Appendix A, λ_em_ = 488 nm (multiline Argon laser).

**Figure 9 pharmaceuticals-15-01172-f009:**
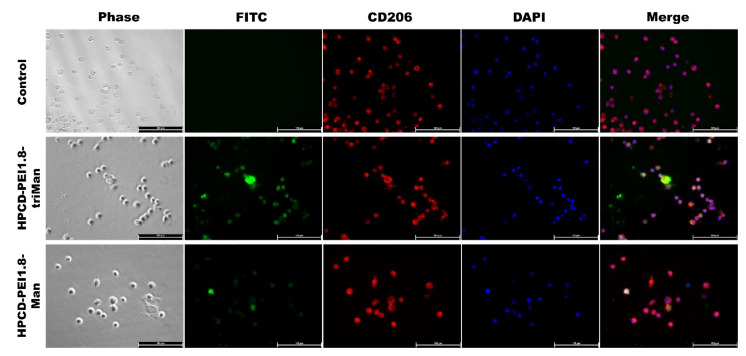
Immunocytochemical evaluation of CD206 in THP-1-derived macrophage-like cells (red channel). Phagocytosis assay with HPCD-PEI1.8-triMan and HPCD-PEI1.8-Man conjugated with a FITC fluorescent label after incubation for 40 min (green channel). Phase contrast microscopy, fluorescent microscopy, blue channel—nuclei stained with DAPI. Scale bar, 100 μm.

**Figure 10 pharmaceuticals-15-01172-f010:**
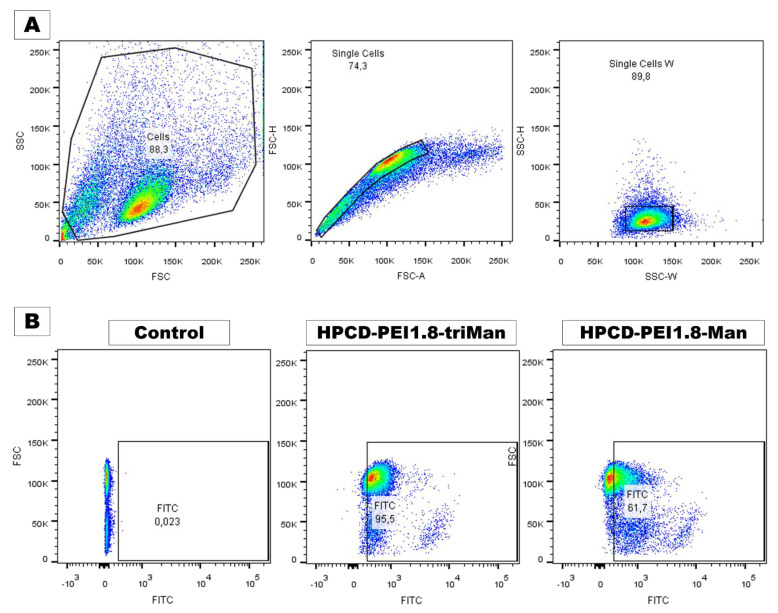
Flow cytometry of macrophage-like cells. (**A**) Gating strategy. Selection of single cells by forward-scattered (single cells) and side-scattered light (single cells W). (**B**) Flow cytometry data on FITC macrophage-fluorescence distribution for cells preincubated with HPCD-PEI1.8-triMan or HPCD-PEI1.8-Man and control cells.

**Table 1 pharmaceuticals-15-01172-t001:** Physico-chemical properties of mannosylated conjugates.

Ligand	Chemical Designation	Molecular Weight, kDa	Molar Ratio *	Hydrodynamic Size **, nm	Zeta Potential **, mV	Polydispersity Index **
**1**	amCD-triMan	2.1	1:1	<5	−2 ± 1	-
**2a**	PEI1.8-triMan	15 ± 5	1:15	6 ± 1	−1.1 ± 0.5	0.30
**2b**	HPCD-PEI1.8-triMan	23 ± 8	5:1:15	17 ± 4	−2.1 ± 0.4	0.28
**3a**	PEI10-triMan	45 ± 10	1:40	8 ± 1	+20 ± 3	0.5
**3b**	HPCD-PEI10-triMan	60 ± 7	10:1:40	48 ± 7	+5 ± 2	0.2

* In the specified order. Molar ratio of the components was calculated based on the integral intensity of the characteristic peaks corresponding to triMan, PEI, and HPCD, in the NMR and FTIR spectra. ** Size distribution values by volume, zeta-potential distribution, and polydispersity index were determined by dynamic light scattering method; 0.01 M PBS (pH 7.4).

**Table 2 pharmaceuticals-15-01172-t002:** ConA-ligand complexation parameters determined by FTIR spectroscopy: complex dissociation constants and stoichiometry. ConA + nLigand ⇌ ConA·nLigand. T = 22 °C. C(Ca^2+^) = C(Mn^2+^) = 1 mM. Phosphate buffer solution pH = 6.4.

Ligand	Chemical Designation	n	−lg *K*_d_ (ConA·nLigand)
**1**	amCD-triMan	1.0 ± 0.2	4.1 ± 0.8
**2a**	PEI1.8-triMan	1.5 ± 0.1	6.5 ± 0.4
**2b**	HPCD-PEI1.8-triMan	1.2 ± 0.2	6.0 ± 0.2
**3a**	PEI10-triMan	1.1 ± 0.2	5.5 ± 0.7
**3b**	HPCD-PEI10-triMan	0.9 ± 0.2	4.4 ± 0.3

**Table 3 pharmaceuticals-15-01172-t003:** Dissociation constants of MF conjugate complexes determined by FTIR spectroscopy. Number of MF molecules held by one conjugate molecule in CD tori (N_CD_) and polymer matrix (N_polymer_). Entrapment efficiency (EE) and loading capacity (LC) of MF by molecular containers. C_ligand_total_ = 0.0001 M, C_MF_total_ = 0.001 M. The conditions are given in the Methods section.

Ligand	−lg *K*_d_ (MF–Ligand)	N_CD_ or N_polymer_	EE, %	LC, %
1	4.0 ± 0.2	1–CD	9 ± 1	20 ± 3
2a	3.3 ± 0.3	(4 ± 1)–polymer	24 ± 3	11 ± 2
3a	3.8 ± 0.4	(17 ± 2)–polymer	84 ± 6	15 ± 3

**Table 4 pharmaceuticals-15-01172-t004:** ^1^ H-chemical shifts corresponding to MF in free form and in the form of non-covalent complexes with ligands 2a and 2b (PEI1.8-triMan and HPCD-PEI1.8-triMan). D_2_O, 400 MHz.

MF Proton Number (Assignment)	^1^ H-Chemical Shifts, ppm		
	MF Free (Figure 6a)	MF + PEI1.8-triMan (Figure 6b)	MF + HPCD-PEI1.8-triMan (Figure 6c)
48 (aromatic)	8.56	8.38	8.51
49 (aromatic)	6.89 and 6.85	7.25 and 7.22	7.30 and 7.27
38 (cyclopropane)	3.34	3.31	3.30
34, 33 (pyrrolidine)	4.08 and 3.76	3.9 and 3.73	3.99 and 3.75 (overlapped)
35, 32 (pyrrolidine)	4.00 and 3.70	3.65 (overlapped) and 3.53	3.88 and 3.51
31 (pyrrolidine–piperidine)	3.89	3.81	3.71
50, 51, 52 (–OCH_3_)	3.50	3.39 (overlapped)	3.45
46, 47 (piperidine)	3.03	2.99	3.01
30 (pyrrolidine–piperidine)	2.75	2.74	2.74 and 2.63
36, 37 (piperidine)	1.84–1.87	1.80 and 1.75	1.81
39, 40 (piperidine)	1.75–1.80	NI *	NI
43, 45 (cyclopropane)	0.99–1.05	0.90	0.78
42, 44 (cyclopropane)	0.88 and 1.21	1.07	1.02–1.04

* Not identified.

**Table 5 pharmaceuticals-15-01172-t005:** Concentrations of MF and EG in solution adsorbed on the cell wall surface and absorbed by *B. subtilis* cells (10^6^ CFU) depending on the composition of the system and incubation time. Incubation in medium LB (pH 7.2), 37 °C. Fluorescent detection of MF and EG: λ_exc_ = 280 nm, λ_emi_(EG) = 320 nm, λ_emi_(MF) = 465 nm–22 °C.

Time	PEI10-triMan 0.5 mg/mL + MF 47 μM			PEI10-triMan 0.5 mg/mL + MF 47 μM + EG 6 mM					
	C in Solution	C Adsorbed on the Cell Wall Surface	C Absorbed Inside the Cell	C in Solution		C Adsorbed on the Cell Wall Surface		C Absorbed Inside the Cell	
	MF, μM			MF, μM	EG, mM	MF, μM	EG, mM	MF, μM	EG, mM
0 min	47	0	0	47	6	0	0	0	0
15 min	29 ± 3	15 ± 3	2.8 ± 0.3	15 ± 2	0.20 ± 0.05	27 ± 3	5.8 ± 0.3	4.4 ± 0.2	0.030 ± 0.005
45 min	15 ± 2	30 ± 4	1.6 ± 0.2	10 ± 1	0.10 ± 0.02	34 ± 4	5.9 ± 0.3	3.3 ± 0.2	0.029 ± 0.006
2 h	10 ± 2	35 ± 4	1.2 ± 0.2	12 ± 1	0.10 ± 0.02	32 ± 4	5.9 ± 0.1	3.1 ± 0.2	0.026 ± 0.004
14 h	12 ± 2	33 ± 4	1.7 ± 0.3	6 ± 1	0.010 ± 0.005	40 ± 5	6.0 ± 0.1	0.5 ± 0.1	0.0010 ± 0.0005

## Data Availability

The data presented in this study are available in the main text and Appendix A.

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
