# Peer review of "Mannosylated Polymeric Ligands for Targeted Delivery of Antibacterials and Their Adjuvants to Macrophages for the Enhancement of the Drug Efficiency"

_pharmaceuticals, 2022, doi:10.3390/ph15101172_

Round 1
Reviewer 1 Report (Previous Reviewer 1)
This revised manuscript provides new cell-based assay data, additional control experiments, and clarifies other points raised during initial review.
However, I want to state unequivocally my opposition to presenting error bars that are chosen arbitrarily and do not reflect the error observed in the data – in this case, as the square root of the average of 3 data points. The whole point of error bars is to represent the error in the measurement being made; choosing to present as an error bar the square root of the mean *does not in any way reflect the variation of the 3 measurements made*. Apparently this choice was made because the standard deviation was perceived to be too small, but that is absolutely not a justification for switching to a metric that is simply not a measurement of error. Either switch to standard deviation, or do what reputable journals have required for years, which is to show the individual 3 measurements so that readers can judge reproducibility for themselves. (btw did the “relative deviation”, however that may be defined, not exceed 10% as in the text, or 15% as in the rebuttal letter?). Bluntly the authors’ response to my initial critique significantly undermines confidence that other studies here are performed and reported in line with typical scientific norms.
The added cell-based data (Fig 9) showing that FITC-labeled ligand is uptaken by CD206+ macrophages adds meaningfully to the manuscript. However, additional controls would be needed to rigorously support the authors’ model: eg, cells that don’t express CD206 *don’t* take up the polymer, since it was designed to interact w CD206? Cells are viable/not permeable after 40min with these concetnrations of polymer?
Overall, the authors want to conclude that the outcome of their distinct vignettes is that they are poised to solve the challenge of antibacterial resistance due reservoirs of bacteria within macrophages. While I agree that some progress is documented here, it’s spread thinly across three relatively distinct lines of investigation, and significant additional work performed with high rigor would be needed to demonstrate that this delivery system (and its adjuvants) provide advantage in targeting MF to macrophages to suppress bacterial infection.
Author Response
Please see the attachment

Reviewer 2 Report (New Reviewer)
The current work focuses on Mannosylated polymeric ligands for targeted delivery of anti bacterials and their adjuvants to macrophages for the enhancement of the drug efficiency. The author’s great effort into the improved manuscript, but minor issues should be addressed.
Introduction
- The introduction provides sufficient background, and all relevant references are included. But, the novelty of this work is not highlighted and it was not clear the author's contribution in comparison to other previous works.
Materials and Methods
- The used reagents with their impurities should be inserted
- Experimental part required rephrasing to be more precise with details and logic to the reader for reproducibility.
- Calculated amount of reagents (gm), dispersed solvent amount (ml), in addition, the concentration should be inserted: not only say HPCD solution in water (0.4 mM) to the CDI solution in DMSO (1.5 mM)
- What is the temperature of the reaction, stirring type and speed of it,…..
- What is the used amount of 0.1-0.01 M HCl
-Condition of measurements is very important to judge the DLS and zeta potential results: What is the condition for measurement of zetapoteantial analysis? Concentration? pH? Solvent used type? One run or multi-run measurements?...
Also the same, what is the condition for measurement of DLS analysis?
Result and discussion
- A zeta potential (ζ) is very important for biomedical applications to achieve expectable and consistent outcomes. The low value of zeta potential implies that the particle may show poor stability in aqueous solutions. Low zeta potential values (0 to ± 5 mV) will improve Van der Waals interparticle attractions and causes rapid coagulation and flocculation of particles. On the other hand, the higher value of zeta potentials implies that the particle may show good stability in aqueous solutions. There is a specific zeta potential value (≈ ± 30 mV) that determines the stability of particles. At this value, high electrostatic repulsive forces between the particles occur.
In this work, no interpretation for Zera potentioal results!! What we can conclude from these numbers!! It's good for bio application or not? Why the sign was negative? Positive?
- Why make IR measurement only until 3000cm-1, where there is an important characterized peak for PEI at 3100-3200 cm-1for NH stretching
- Please make one system legend in all figures, sometimes Time (h) and another make Time, h
And C, mg/mL or C (mg/mL)
Conclusion
The conclusion is very long; please rephrase it to be shorter with the main target and outputs
E.g. lines 802-808, all this general information could be deleted.
References
- Correct the reference with complete citation e.g. no.7, 10, 18, 24.
- make one system style and insert the more recent related citation.
Author Response
Please see the attachment

This manuscript is a resubmission of an earlier submission. The following is a list of the peer review reports and author responses from that submission.
Round 1
Reviewer 1 Report
This manuscript by Zlotnikov et al seeks to improve antibiotic efficacy by creating complex conjugate systems that may expedite delivery of fluoroquinolones to macrophages. Additionally these complexes can be loaded with Moxofloxacin which may prolong this agent’s antibacterial activity. Moreover use of adjuvants may enhance the permeability of bacteria to doxycycline.
A central concern is that the multiple threads of this manuscript are not sufficiently related as to form a coherent story. While eradication of microbes within macrophages may be an important goal, no evidence is provided that the complex conjugates presented here are actually able to deliver agents to macrophages, and the binding studies use ConA rather than the mannose receptor intended for targeting to macrophages, CD206. It’s thus unclear to what extent this system represents a promising starting point for solving the challenge of dormancy in macrophages.
A separate line of experiments sought to assess whether binding MF within these mannosylated conjugates may improve its antibiotic efficacy. While it does seem that MF is sequestered by the cyclodextrin portion, the data in Fig. 7 raises many questions. How does viability rebound so strikingly for MF treated cells between 24 and 48h? This type of ‘catching up’ only seems possible if the MF bacteria resume growth while control bacteria have reached stationary phase, which doesn’t seem like a fair measurement. Additionally, the addition of eugenol, menthol, and safrole is uninterpretable without data indicating the effect of these molecules as single agents.
The final line of experiments shifts to monitoring the uptake of antibiotics into bacteria. Here doxycycline is used, again making this vignette feel disconnected from previous vignettes. Again there are issues with appropriate controls, eg, why is it not important to demonstrate that the SDS wash used to remove eugenol doesn’t influence absorption of Dox? (The text parenthetical that a control experiment “had no effect on membranes” is vague and does not substitute for data).
Overall, this manuscript seeks to combine three tangentially related vignettes, each of which in my view make incremental advances toward the goal of improved antibiotic efficacy.
IMPORTANT QUESTION: The error bars in Fig 7 appear questionable. Every condition at time zero had an error bar (undefined – std dev?) of 10%? And every control condition (orange bar) at every time point also had an error bar of 10%? To my eye, the error bar appears for every bar to be directly proportional to bar height and to be ca. 10% of the bar height. It gives the impression that error was arbitrarily set to 10%. Notably this is not the case in Fig S1. I hope the editorial staff will be very mindful of this observation and that a very clear explanation is provided prior to publication anywhere.
Reviewer 2 Report
This manuscript is a study of the application of mannose clusters to DDS. This field is important because of the great demand for research in this area.
 Various assays are being performed, but the most important is the synthesized chemicals. There is a lack of reliable data on synthesis and purity of the substance. comparison of IR peaks is not enough to determine the purity of the synthesis. data is needed, such as whether the TLC peak is a single spot. 1H-NMR should be also sufficient data if properly analyzed, but assignment is not appropriate. Reference 24 is on synthesis but is not adequate.
 This manuscript is not worthy of acceptance. I recommend rejection.
Reviewer 3 Report
The manuscript “Mannosylated polymeric ligands for targeted delivery of anti-bacterials and their adjuvants to macrophages for the enhancement of the drug efficiency” researched three–component polymer system for delivering fluoroquinolones to macrophages and antibacterial activity. I think the author pay more attention to the work, the data was reasonable and enough to discuss the results. I have some comments as follows:
1. How the antibacterial property of drug loading system ligand 2b & 3b, could they compare with ligand 2 a 2b and 1?
2. In figure 7 and S4 why use menthol adjuvants?